# DFT Study of Regio- and Stereoselective 13DC Reaction between Diazopropane and Substituted Chalcone Derivatives: Molecular Docking of Novel Pyrazole Derivatives as Anti-Alzheimer’s Agents

**DOI:** 10.3390/molecules28041899

**Published:** 2023-02-16

**Authors:** Sadeq M. Al-Hazmy, Mohamed Oussama Zouaghi, Nasser Amri, Youssef Arfaoui, Ibrahim A. Alhagri, Naceur Hamdi

**Affiliations:** 1Chemistry Department, College of Science, Qassim University, Buraidah 51452, Saudi Arabia; 2Laboratory of Characterizations, Applications and Modeling of Materials (LR18ES08), Department of Chemistry, Faculty of Sciences, University of Tunis El Manar, Tunis 2092, Tunisia; 3Department of Chemistry, Faculty of Science, Jazan University, P.O. Box 2097, Jazan 45142, Saudi Arabia; 4Department of Chemistry, College of Science, Ibb University, Ibb P.O. Box 70270, Yemen; 5Department of Chemistry, College of Science and Arts, Qassim University, Ar Rass 52719, Saudi Arabia

**Keywords:** 32CA, pyrazoles, oxadiazoles, diazopropane, DFT, molecular docking

## Abstract

In the present work, a combination of experimental and density functional theory (DFT) investigation of the (3+2) cycloaddition reactions of diazopropane with chalcone derivatives was reported. All calculations were performed using several DFT approaches (B3LYP, M06, M06-2X) and 6-311+G(d, p) basis set. Based on the NMR, MS analyses and IRC calculations, the pyrazole derivatives are the kinetic adducts over the oxadiazoles. The use of two equivalents of diazopropane leads to thermodynamical products. A molecular docking analysis was performed to investigate the efficiency of the obtained products against selected drug targets in anti-Alzheimer ligand-receptor interactions. We revealed that the ligands selected were bound mainly to the catalytic (CAS) and peripheral (PAS) anionic sites of acetylcholinesterase (AChE) and butyrylcholinesterase (BuChE) inhibitors, respectively. The selected ligands 1, 3, 4 and P14 may act as the best inhibitors against Alzheimer’s disease (AD).

## 1. Introduction

The majority of heterocyclic compounds exhibit various biological activities, allowing them to act as anti-inflammatory, anticancer, antidiabetic, and antipsychotic agents [1] and are present in most pharmaceuticals currently marketed [2]. Quinoline, pyrazole, oxadiazole, thiophene, diazepine and others were the most studied heterocyclic compounds in organic synthesis and drug discovery [3,4,5]. Taking the case of pyrazole-based pharmaceuticals, they treat a wide array of diseases and conditions including obesity [6,7], diabetes [8], cancer [9], microbial [10] and viral infections [11], pain and inflammation [12] and many neurological disorders [13]. As well, the oxadiazole derivatives have attracted a wide attention of chemists in search for new therapeutic molecules and constitute an important class of compounds with a wide range of biological properties. The oxadiazoles and their derivatives were reported to possess antimicrobial [14], analgesic [15], anti-inflammatory [15], antibacterial [15], anticancer [16] and antifungal [17] activities. Recently, novel oxadiazole derivatives were discovered by S. K Verma et al. [18] and V. B. Makane et al. [19] as antitubercular agents with limited activity against drug-resistant tuberculosis. The oxadiazole and pyrazole derivatives represent an important class of heterocyclic compounds. They can be obtained by different methods such as cyclocondensation [20,21,22] or multicomponent reactions [23,24,25,26]. The [3+2] cycloaddition reactions 32CA are efficient way to synthesize the pyrazole and oxadiazole derivatives [27,28,29]. Furthermore, the 32CA reaction is one of the most useful reactions in organic synthesis [30]. The concept of 32CA was described by Rolf Huisgen 60 years ago (nowadays known as “Huisgen reactions” or “Huisgen chemistry”) [31]. 32CA reaction possesses three types of mechanisms, (*i*) non-polar mechanisms when it happens a conversion into adducts may result in biradical intermediates [32], (*ii*) polar mechanisms [33,34], (*iii*) zwitterionic or biradical adducts with “extended conformation” may be exist in the reaction environment independently of [3+2] cycloadducts [35]. In the last 10–15 years, many examples of experimental studies on the 32CA between diaryl and aryl-alkyl nitrones with pi-deficient alkenes were described in the literature [36,37]. The procedure of the 32CA of DP with DAP is illustrated in Figure 1.

A number of theoretical and experimental studies have shed light on the mechanism of reaction of Huisgen chemistry [38,39]. Most of the experimental 32CA reactions are largely based on diazoalkane to functional alkynes [40], with several cyclobutenes [41], imidates [42], and propargyl alcohols [40]. In 2003 [43], we showed a simple method for the 32CA based on the reaction the 2-Diazopropane (DAP) DE with enones (e.g., chalcone derivatives) DP. In the present work, we report a study of regio- and stereoselective 32CA reactions of diazopropane DAP with dipolarophile DP by several theoretical approaches, namely, DFT-based reactivity indices, transition state (TS) and (IRC) calculations. We have also elucidated the biological activities of the most stable studied products, using the molecular docking approach, which predicts the preferred orientation, affinity, and interaction between selected ligands in the binding site with different proteins. Both DAP and DP molecules are sketched in Figure 2.

## 2. Results and Discussion

### 2.1. Topological Analysis of the 1,3-Dipolar Cycloaddition Reaction

As a starting point, we studied the 1,3-dipolar cycloaddition reaction between DAP and DP. In this study, we used (*E*)-3-(4-methoxyphenyl)-1-phenylprop-2-en-1-one (1H) as the dipolarophile. Analogously, for each 1,3-dipolar cycloaddition, two regiospecific cycloadducts can be formed: the 1,2-adduct with DAP to the ketone group and the 3,4-adduct with DAP to the second double bond alkene (Figure 3).

### 2.2. Application of the Rule of Houk and Local Regioselectivity Analysis

In this approach, we applied the rule of Houk [44,45], for the heterocycloaddition reaction. The rule of Houk, “small-small” and “large-large” type interactions are more favored in comparison with those of ‘large-small’ and ‘small-large’ types as explained in Figure 4. Following the results in Figure 5, the reaction occurs for the first time between N35 and C1 in the case of P1 (DAP and (C=C) alkene as DP). Then, we can conclude that the first formed bond length during the reaction is N35-C1. However, in the case of the C=O alkene as DP, we have found less favored interactions (the two opposed sites, which are concerned by the interaction, possess large/small Molecular Orbital Coefficients, MOC). These results are in good agreement with the experiment [43], and we can understand that the cycloaddition occurs between the DAP and the C=C alkene.

To more investigate thoroughly the local reactivity of the sites, which are concerned by the 32CA reaction, we have calculated the local Parr reactivity descriptors. (Table 1)

According to the data in Table 1, we have found that mostly N35 shows the highest $P_{k}^{-}$ value and, consequently, the most nucleophilic site. In addition, we have obtained that C1 possesses the highest $P_{k}^{+}$ value and, consequently, the most electrophilic site. We can conclude that the cycloaddition with the C=C alkene is the most favored and this is in good agreement with the investigation of the Houk rule application.

### 2.3. Analysis of Transition States TS and Intrinsic Reaction Coordinate IRC

In order to investigate the effects of halogen, multi fluorine and the nature of DP(C=C or C=O) on the kinetic and thermodynamic profiles for the different desired products, we have calculated the transition states of the different pathway reactions followed by IRC calculations. These latter were also computed to check the reliability of the obtained TSs for the reactants and products. Several hybrid (B3LYP), meta-hybrid (M06) and double-meta hybrid (M06-2X) functionals were tested. The IRC plots are sketched in Appendix A.

Following the data in Appendix A, the B3LYP gave the most suitable IRC, unlike M06 and M06-2X, which show that the product was unstable. The M06 and M06-2X methods showed poor results and were far from the experiment. Afterward, we have continued the calculations in this study at the B3LYP level. In Appendix A, IRC plots as a function of dispersion (B3LYP-d3) and solvation model are presented. Taking into account the functional dispersion and solvents effects has a very slight impact on the $E_{a}$ and $\Delta_{r}E$ values.

The computed energy values of the different TSs and products with respect to the reagent were summarized in the energy diagram (Figure 6). We have found that 1HP1 (pyrazole) was the most stable product. The latter was obtained by the attack on the C=C bond. However, 1HP3 and 1HP4 (oxadiazoles), which are obtained after the reaction on the C=O bond were kinetically and thermodynamically unfavoured since they exhibited the highest activation energy $E_{a}$ ($E_{a}\;$= E(TS) − E(Reagent)) and free energy of reaction $\Delta E_{r}$ ($\Delta E_{r}$= E(Product) − E(Reagent)), respectively. Kinetically, the pyrazoles 1HP1 and 1HP2 possess the lowest activation energy values and are also, thermodynamically, the most stable ones. Consequently, we have investigated the reaction of the obtained pyrazoles 1HP1 and 1HP2 with a second equivalent of 2-diazopropane. The component 1HP14 was the thermodynamic product since it exhibits the lowest free energy value.

The effects of halogen, multi-fluorine and the nature of DP(C=C or C=O) on $E_{a}$ and $\Delta E_{r}$ values and, consequently on the kinetic and thermodynamic profiles were investigated. The obtained results are presented in the Table 2 and Table 3. The four TSs(TS1HP1-4) including all the standard data required in computational works are included in the Appendix A.

According to the data in Table 3, the compound **9** (1F(m): mono-fluorine in meta position compound) was the kinetic product since it exhibits the lowest activation energy value ($E_{a}=12.05\;\mathrm{kcal}/\mathrm{mol}$). In the case of the compound **12** (3F(p-o): having three fluorides in *para* and *ortho* positions) was the most stable product since it shows the lowest free energy change value ($\Delta E_{r}=-10.67\;\mathrm{kcal}/\mathrm{mol}$). The formation of the adduct 6 (1F(p)P3) may be thermodynamically unstable since it possesses the highest activation energy value ($E_{a}=25.55\;\mathrm{kcal}/\mathrm{mol}$). As main results, we have found that the [3+2] cycloaddition reaction between the DAP and the C=C DP was the most favored, and this is in accordance with the local reactivity investigation. The modeling of the chalcone’s aromatic ring by the insertion of one fluorine at the *meta* position leads to the kinetic product, while the electrophilic substitution by 3 fluorides at *para* and *ortho* sites gives the thermodynamic product. Furthermore, the obtained IRC data proved that the reaction with C=O alkene is kinetically very difficult (Appendix A).

In order to evaluate the polar nature of these 32CA reactions, the Global Electron Density Transfer (GEDT) was analyzed (Table 4). According to the analysis of GEDT values in Table 4, we have found that all studied reactions are defined as polar 1,3-dipolar cycloadditions. The charge transfer (CT) within reagent becomes more important when the cycloaddition occurs between DAP and the C=O DP (as  DP). In addition, the CT increases as a function of the number of fluorines. At another level, a new N35-C1 σ-bond appears before the C_2_-C_34_ bond in the case of the reaction between DAP and the C=C alkene. Except in the case of 1F(p)P2, we have found close L values. The fast formation of the N-C bond is in good agreement with the investigation of the rule of Houk application. The studied reactions are under orbital control.

### 2.4. Molecular Docking Analysis

In recent years, molecular docking simulations have become a key tool in computer-assisted drug design to analyze ligand inhibitor behavior against selected proteins. The docking investigation provides a detailed analysis of the interactions between the ligands 1HP1-4, 1HP14, and 1HP24 and the amino acid proteins. The binding affinity scores for inhibitors were illustrated in Figure 7, Figure 8, Figure 9, Figure 10, Appendix A and Table 5.

#### 2.4.1. Molecular Docking Results for Human Serum Albumin (HSA)

Based on the best binding free energies and binding modes, the possible HAS-ligand interactions are depicted below in Figure 7 and Appendix A. Greater negative values indicate stronger interactions between the ligand and the receptor, hence better binding affinity [46]. The (1HP1-4, 1HP14, and 1HP24) ligands with HAS showed higher negative binding energy. Furthermore, ligand 1HP4 (Appendix A) exhibited notably, the lowest binding affinity value (−8.6 kcal/mol).

The components 1HP1-4, 1HP14, and 1HP24 formed hydrogen bonds with Lys195, Lys413, Arg209, Arg117, Arg218, Arg222, Cys448, Asp451, and Trp214 at a distance of 3.36, 3.02, 4.77, 3.25, 3.17, 2.98, 3.13, 3.77 and 3.29 Å, respectively. Moreover, Trp214, Val409, Leu115, and Ala213 interacted via pi-sigma forces. The details of the interactions are summarized in Figure 7 and Appendix A.

**Figure 7 molecules-28-01899-f007:**
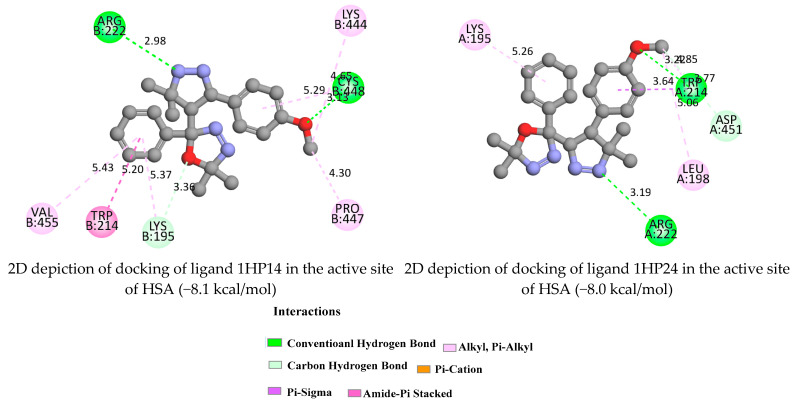
2D depiction of docking of ligands 1HP14 and 1HP24 in the active site of Human Serum Albumin.

#### 2.4.2. Molecular Docking Results for Hydrolase (AChE and BuChE)

The objective of this process was to visualize the interaction between the selected ligands 1HP1-4 and the receptor proteins Acetylcholine (AChE) and Butyrylcholinesterase (BuChE). The molecular docking predicted that these compounds predictively bound the peripheral (PAS) anionic site of AChE and the catalytic (CAS) anionic site of BuChE.

In the human Acetylcholine (hAChE), PAS region was in contact with TRP286, TYR341, TYR124, ASP74, VAL365, TYR72, THR75 and LEU289 residues and TRP82, TYR332 and ASP70 in hBuChE [47,48]. The catalytic active site (CAS) region is constituted by a catalytic triad: Ser203, His447, and Glu202 in human AChE (hAChE) and Ser198, His438, and Glu197 in human BChE (hBuChE) [49]. Trp279 and Tyr70 contribute to the PAS at the mouth of the gorge (Eichler et al., 1994) [50].

Regarding AChE (Figure 8 and Appendix A), docking results revealed the Pi-Pi Stacking, Pi-Sigma, Pi-Cation, Pi-Pi T-Shaped, Hydrogen Bond and, Pi-donor Hydrogen Bond. Taking the case of component 1HP4, docking results revealed strongly core and key residues TYR70, TYR121, TYR334, PHE288 and TRP279 in the peripheral active site (PAS). Furthermore, all selected ligands 1HP1-4 produced hydrogen bond interactions with Arg517 (ligand 1), VAL518, ARG515 (ligand 2), ASN424 and ARG517 (ligand 3) and PHE288 and TYR70 (ligand 1HP4).

Molecular modelling studies were performed to explore the possible mode of interactions between the most active compounds 1HP1-4 and BuChE. The most potent BuChE inhibitor showed interactions with TRP82, HIS438, TRP430, TYR440, and TYR332 residues of the human butyrylcholinesterase enzyme (hBuChE).

The compounds 1HP1, 1HP3 and 1HP4 could also interact with both the peripheral (PAS) and the catalytic (CAS) active site regions, with higher affinity for the CAS compared to the PAS cavity, which confirms their high-inhibitory potency. TYR 332, TYR128 and TRP82 from the active pocket of the protein were observed to form hydrogen bonds. The fluorine group in the para-positions on phenyl rings created a halogen interaction with the GLY115 residue. The highest binding affinity was achieved for all ligands 1HP1-4. The ligands 1HP3 and 1HP4 exhibited the highest binding affinity (−10.2 and −10.5 kcal/mol, respectively). The CAS and PAS were clearly explained in Figure 9, Figure 10, Appendix A).

**Figure 8 molecules-28-01899-f008:**
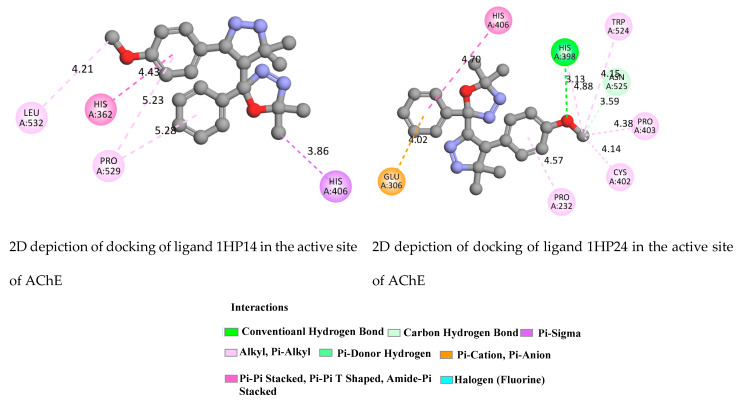
2D depiction of docking of ligands 1HP14 and 1HP24 in the active site of AChE (PDB ID: 1ACJ).

**Figure 9 molecules-28-01899-f009:**
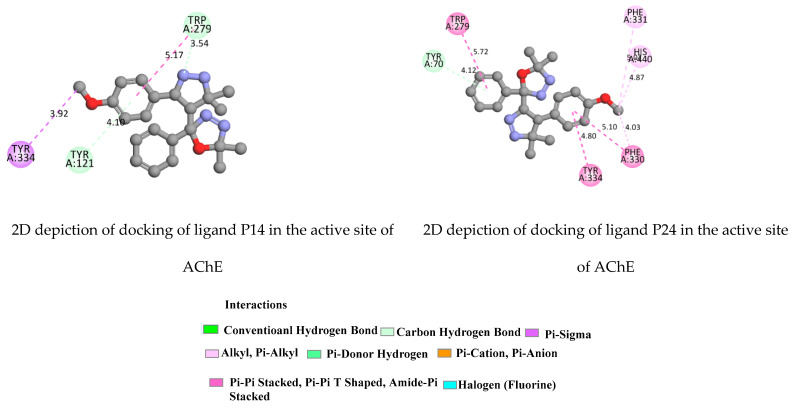
2D depiction of docking of ligands 1HP14 and 1HP24 in the active site of AChE (PDB ID: 1EVE).

**Figure 10 molecules-28-01899-f010:**
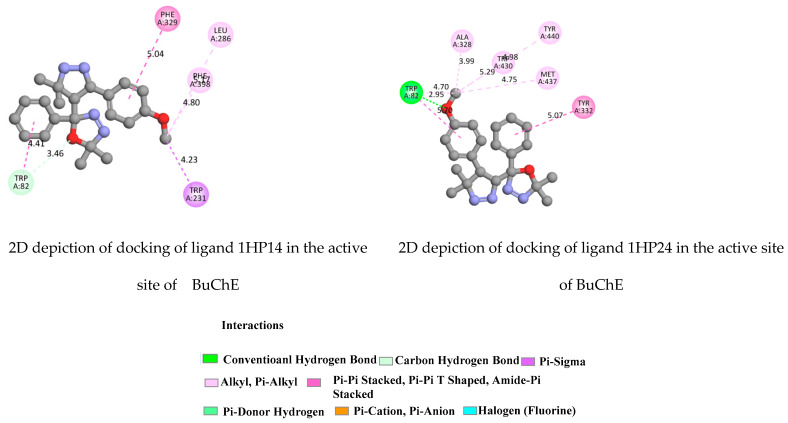
2D depiction of docking of ligands 1HP14 and 1HP24 in the active site of BuChE (PDB ID: 1P0I).

## 3. Materials and Methods

### 3.1. Methodological and Computational Details

One of the main purposes of this work is to provide an explanation of the 13DC regioselectivity. Eisenstein, et al. [51] have claimed complete correspondence of observed cycloaddition regioselectivity to frontier orbital theoretical predictions using Hückel calculations. The quantum chemical calculations were carried out at DFT using Gaussian 16A suite of programs [52]. The geometries of all reactants and products have been optimized at 6-311G+(d, p) [53] basis set with a collection a hybrid (B3LYP) [54] and meta-hybrid (M06 and M06-2X) [55] functionals in the gas phase. The Berny algorithm [56] was used to locate the transition states (TSs) for the 13DC reaction. The intrinsic reaction coordinate (IRC) path was traced to check the energy profiles that connect each TS with the two corresponding minima of the proposed mechanism [57,58]. In the all studied TS structures, one new σ bond was formed and was attributed to the bond between the strongly nucleophilic atom (X) of the DAP moiety and the strongly electrophilic atom (Y) of the dipolarophile. The index value of l was expressed as:(1)$$l=1-\left(\frac{r_{X-Y}^{TS}-r_{X-Y}^{P}}{r_{X-Y}^{P}}\right)$$
where $r_{X-Y}^{TS}$ is the distance between the reaction centers *X* and *Y* in the transition structure and $r_{X-Y}^{P}$ the same distance in the corresponding product [32].

Local reactivity index, named Parr function P(r) [59], which is obtained from the ASD at the radical cation and at the radical anion of the corresponding reagents, which is given by the following equations,
(2)$$P^{-}(r)=\rho_{S}^{rc}(r)\;\mathrm{for}\;\mathrm{electrophilic}\;\mathrm{attacks}$$
and
(3)$$P^{+}(r)=\rho_{S}^{ra}(r)\;\mathrm{for}\;\mathrm{nucleophilic}\;\mathrm{attacks}$$
where $\rho_{S}^{rc}$(r) is the ASD of the radical cation, and $\rho_{S}^{ra}$(r) is the ASD of the radical anion. Each ASD condensed at the different atoms of the radical cation and radical anion provides our local nucleophilic $P^{-}$(r) and electrophilic $P^{+}$(r) Parr functions of the neutral system.

In the polar 32CA reactions, the electron density demanded for the formation of the pseudoradical centers is reached mainly by the depopulation of the C=C double bonds present in the reagents. This electron density mainly comes from the Global Electron Density Transfer GEDT that takes place along the polar process [57,60]. Thus, the pseudoradical center created in the electrophilic species was formed mainly in the most electrophilic center of the molecule, which is the center with the highest spin density achieved through the GEDT process.

### 3.2. Selection of Inhibitors

The docking study was performed using ezCADD [61] Smina [62] software (https://www.dxulab.org/software) (accessed on 1 August 2022). The ezCADD is a web-based CADD modeling environment where one of its applications is to simulate protein-ligand interactions. 2D/3D depictions were visualized using the BIOVIA discovery studio 2021 client software [63]. The simulation system was built on the crystal structures of PDB ID: 1AO6 [64], 1P0I [65], 1ACJ [66] and 1EVE [67], which were downloaded from the Protein Data Bank (https://www.rcsb.org (accessed on 1 August 2022)). These proteins were classified into antibodies against Human Serum Albumin (HAS) and Hydrolase (hydrolase carboxylic esterase and hydrolase inhibitor). The water molecules and ligands were removed from the PDB file using pymol [68] and polar hydrogen bonds were added [69]. Automatic cavity detection was performed using *fpocket3* [70]. The minimum binding affinity values (kcal/mol) were calculated to evaluate the interactions between the docked ligands and the studied proteins.

### 3.3. Synthesis of Pyrazole Derivatives (D1–D3)

Figure 1 depicts a probable mechanism for the reaction involving the Michäel acceptor and one or two diazopropane molecules. Melting points were measured with a Büchi-510 capillary apparatus. NMR spectra were recorded on a Brücker AC-300 (1H at 300 MHz and 13C at 75 MHz). The chemical shifts (in ppm) are counted positively towards the weak fields with respect to the tetramethylsilane (TMS) taken as internal reference. The FTIR spectra were recorded on a spectrometer of type Perkin- Elmer 298.The mass spectra are measured thanks to a Hewlett Packard 5890 and 5971 series II apparatus (C.P.G.-mass). For the mass spectra, the results are given in *m*/*z* value for the most important peaks with their relative abundance. All the spectra were realized by electronic impact (EI). Thin layer chromatography was performed on commercial plates: Schleicher and Schuell, silica gel (reference 394732) or Merck. The products were purified by chromatography on silica column; the mode of filling of the column and the operating mode are those described by Taber et al. [71] and Still et al. [72]. The silica used is of Merck origin. The mass spectrum MS of pyrazoline D2 is included in Appendix A, the NMR spectra are illustrated in Appendix A and HMBC of the titled product in Appendix A. The found melting point values in the case of D1Cl are close with those reported by N. Hamdi et al. (2002) [73]. The origin of the used organic reagents and solvents is sigma aldrich. Bolte et al. (2009) have reported the synthesis of the used methoxychalcone including the crystallographic and experimental details of the product [74].

#### 3.3.1. Synthesis of 2-Diazopropane

The synthesis of 2-diazopropane was reported by D. E. Applequist and H. Babad in 1962 [75]. 2-Diazopropane is very unstable and never isolated in the pure state. The DAP in solution is prepared for less than two hours and stored at −60 °C. This 2-Diazopropane solution is added with a syringe in small fractions, quickly, in order to avoid their heating. All these addition reactions are done in Erlenmeyer flasks, with stirring and cooling during the reaction. Remember that 2-Diazopropane decomposes when the temperature increases; this decomposition, competing with cycloaddition reactions, requires the use of an excess of 2-Diazopropane that is all the greater as the reaction is slow [43].

#### 3.3.2. Synthesis of 4-p-Methoxyphenyl-5,5-dimethyl-3-benzoyl-4,5-dihydro-1H-pyrazole (D1H)

Yield = 95%. Yellow crystals. Melting temperature = 151 °C, FTIR ${\bar{\nu }}_{cm^{-1}}$: 3393 (NH), 1636 (C=O), 1532 (C=N). ^1^H NMR (300 MHz, CDCl_3_) δ ppm: 0.92 (s,3H, CH_3_(a)); 1.39 (s,3H, CH_3_(b)); 3.85 (s,3H, O-CH_3_); 4,. 5 (s,1H, H4); 6.82 (s, 1H, NH); 6.80–8.2 (Harom.). ^13^C NMR (75 MHz, CDCl_3_) δ ppm: 22.4(CH_3_(a)); 29.3 (CH_3_(b)); 55.8 (O-CH_3_); 152.4 (C_3_); 57.2 (C_4_); 67.7 (C_5_). 188.0(C=O); C(aromatic) 110–140. MS (EI): (int.rel. %: 308 ([M+]–N2), 12%).

#### 3.3.3. Synthesis of 4-p-Methoxyphenyl-5,5-dimethyl-3- p-bromobenzoyl-4,5-dihydro-1Hpyrazole (D1Br)

Yield = 65% Yellow crystals. Metlting temperature = 165 °C. FTIR, ${\bar{\nu }}_{cm^{-1}}$: 3681 (NH), 1657 (C=O), 1597 (C=N). MS (EI): (int.rel. %: 386 ([M+], 12%). ^1^HNMR (300 MHz, CDCl_3_) δ ppm: 0.85 (s,3H, CH_3_(a)); 1.33 (s,3H, CH_3_(b)); 3.79 (s,3H, O-CH_3_); 4.65 (s,1H, H4); 6.1 (s, 1H, NH); 6.70–7.98 (Harom.). ^13^C NMR (75 MHz, CDCl_3_) δ ppm: 22.4 (CH_3_(a)); 28.9 (CH_3_(b)); 55.8 (O-CH_3_); 157.5 (C3); 56.8 (C4); 68.7 (C5); 189.1 (C=O).

#### 3.3.4. Synthesis of 4-p-Methoxyphenyl-5,5-dimethyl-3-p-chorobenzoyl-4,5-dihydro-1Hpyrazole (D1Cl)

Yield = 90% (Yellow crystals). Melting temperature = 145 °C. FTIR, ${\bar{\nu }}_{cm^{-1}}$: 3300 (NH), 1635 (C=O), 1550 (C=N). SM (EI): (int.rel. %: 342 ([M+], 16.9%). ^1^H NMR (300 MHz, CDCl_3_) δ ppm: 0.89 (s,3H, CH_3_(a)); 1.3 (s,3H, CH_3_(b)); 3.72 (s,3H, O-CH_3_); 4.02 (s,1H, H4); 6.18 (s, 1H, NH); 6.72–7.42 (Harom.). ^13^C NMR (75 MHz, CDCl_3_) δ ppm: 22.8 (CH_3_(a)); 29.3 (CH_3_(b)); 55.5 (O-CH_3_); 154.5 (C3); 57.4 (C4); 69.1 (C5); 189.3 (C=O).

#### 3.3.5. Oxidation of ∆^2^ − Pyrazolines (D1)

In a 250 mL bicol flask fitted with a stirrer, bromine bulb and refrigerator topped with a CaCl_2_ guard tube, 40 g of MnO_2_ is suspended in 100 mL of anhydrous dichloromethane. The mixture is degassed with argon for 10 min. 205 mmol of ∆^2^ − pyrazoline D2(H, F, Cl and Br) in solution in 100 mL of anhydrous dichloromethane is added dropwise and left under effective stirring for 30 min. The solvent is filtered through sintered glass (n°4) and removed by a rotary evaporator. The product D3(H, F, Cl and Br) obtained is purified on silica column eluted with petroleum ether enriched to 20% ethyl acetate [43].

#### 3.3.6. Synthesis of 4-p-Methoxyphenyl-3,3-dimethyl-5-p-bromobenzoyl-3H–pyrazole (D2Br)

Yield = 35%. Melting temperature = 117 °C. IR ${\bar{\nu }}_{cm^{-1}}$: 1664 (C=O), 1602 (C=C-N=N). MS (EI): (int.rel. %: 328([M+], 49%). 1HNMR (300 MHz, CDCl_3_) δ ppm: 1.81 (s, 6H, CH_3_(a,b).), 3.82 (s, 3H, OCH_3_), 6.88–7.54 (Harom). ^13^C-NMR (75 MHz, CDCl_3_) δppm: 24.6 (CH_3_(a),(b); 55.3(O-CH3); 110.4(C3), 155.6 (C4), 145.12 (C5), 196.0 (C=O).

#### 3.3.7. Synthesis of 4-p-Methoxyphenyl-3,3-dimethyl-5-p-chlorobenzoyl-3H–pyrazole (D2Cl)

Yield = 60%. Melting temperature = 96 °C. IR ${\bar{\nu }}_{cm^{-1}}$: 1630 (C=O), 1520 (C=C- N=N). MS (EI): (int.rel. %: 312([M+]-N2), 21 %). ^1^H NMR (300 MHz, CDCl_3_) δppm: 1.30 (s, 6H, CH_3_(a,b).), 3.81 (s, 3H, OCH_3_), 6.70–7.06 (Harom). ^13^C-NMR (75 MHz, CDCl_3_) δppm: 29.7 (CH_3_(a),(b); 58.8 (O-CH_3_); 111.1 (C3), 165.4 (C4), 149.4 (C5), 190.2 (C=O).

#### 3.3.8. Synthesis of the 4-p-Methoxyphenyl-3,3-dimethyl-5-benzoyl-3H–pyrazole (D2H)

Yield = 25%. Melting temperature = 89 °C. IR $\bar{\nu }$: 1625 (C=O), 1510 (C=C-N=N). MS (EI): (int.rel. %: 324 ([M+], 16 %). ^1^H NMR (300 MHz, CDCl_3_) δppm: 1.78 (s, 6H, CH_3_(a,b).), 3.82 (s, 3H, OCH_3_), 6.80–7.80 (Harom). ^13^C-NMR (75 MHz, CDCl_3_) δppm: 24.9 (CH_3_(a),(b); 55.7 (O-CH_3_); 111.1 (C3), 165.0 (C4), 156.0 (C5), 206.1 (C=O).

#### 3.3.9. Synthesis of the 2-(4-p-Methoxyphenyl-5,5-dimethyl-4,5-dihydro-3H-pyrazol-3-yl)-5,5-dimethyl-2′-benzoyl-2′,5′-dihydro-1,3,4-oxadiazole (D3)

A solution of 1.5 mmol of enone (chalcone derivative) in 20 mL of anhydrous dichloromethane is cooled to −60 °C and added fractionally to 10 mL of freshly prepared 2.7 M DAP solution stored at (−60 °C). The very slow decoloration of DAP indicates a slowing down of the reaction [73,74]. After one night at 0 °C, the solvents are evaporated at low temperature, the obtained oil is chromatographied on 150 g of silica by eluting with hexane progressively enriched to 20% of ethyl acetate. The majority pyrazoline D2 is recovered, followed by the more polar adduct D3 [75].

Yield = 25%. IR ${\bar{\nu }}_{cm^{-1}}$: 980–1075 (C-O),1520 (N=N). MS (EI): (int.rel. %: 400 [M-N2] +, 16%). ^1^H NMR (300 MHz, CDCl_3_) δppm: 0.72(s, 6H, CH_3_(a′), 1.4(s, 6H, CH_3_(b′), 1.3(s, 3H,CH_3_(c),(d), 1.5(s, 3H,CH_3_(c),(d), 3.82 (s, 3H, OCH_3_), 6.5–7.5(Harom). ^13^C-NMR (75 MHz, CDCl_3_) δ ppm: 21.8 CH_3_(a′), 25.3 CH_3_(b′), 24.5 CH_3_(c),(d), 26.7 CH_3_(c),(d), 55.7 (O-CH_3_); 90.9(C5′), 94.7 (C3′), 123.4 (C2).

## 4. Conclusions

We have presented a theoretical study of the regio- and stereoselectivity of the [3+2] cycloaddition reactions between 2-diazopropane DAP and chalcone derivatives, as well as experimental results. The experimental organic syntheses of D1 pyrazoles were confirmed and characterized by ^1^H NMR, ^13^C NMR and SM techniques. The global reactivity allowed us to confirm that the 13DC reaction occurs according to normal electron demand. In addition, we have found that the N35-C1 appears first during the reaction, and consequently, the formation of pyrazole derivatives is favored over oxadiazoles. This outcome is in good agreement with the l  calculated values. The analysis of GEDT values showed that the reactions are polar. The polarity increases as a function of the number of fluorides. The impact of halogens (F, Br, and Cl) on the kinetic and thermodynamic profiles was investigated. We have obtained that fluorine gives the lowest activation energy and, consequently, the kinetic product and, thermodynamically, the most stable, which is attributed to the preference of electronegative fluorine to be attached to a more highly alkylated and therefore more electropositive framework. The use of one fluorine at the *para* position on the 1,3-dipolar cycloaddition reactions increases the activation barriers and increases the stability of the product. Contrary to The use of one fluorine at the *meta* position decreases the activation barriers. The presence of three fluorides at *para* and ortho positions in the 13DC reaction decreases the activation barriers and the stability of the product. In addition to the stability of the pyrazole derivatives versus the oxadiazole ones, the use of two equiv of DAP leads to the formation of two thermodynamic products (1HP14 and 1HP24). The Docking simulations showed that ligands 1HP1, 1HP3 and 1HP4 were bound mainly to the CAS and PAS of AChE and BuChE inhibitors, respectively. The compound 1HP4 was found to be the most effective inhibitor against BuChE.

## Data Availability

The datasets generated during and/or analyzed during the current study are available from the corresponding author on reasonable request.

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
