# Peer review of "DFT Study of Regio- and Stereoselective 13DC Reaction between Diazopropane and Substituted Chalcone Derivatives: Molecular Docking of Novel Pyrazole Derivatives as Anti-Alzheimer’s Agents"

_molecules, 2023, doi:10.3390/molecules28041899_

Round 1
Reviewer 1 Report
1. The topic should be changed: «:» should be replaced with «and». Or any other ideas. But «:» is not applicable for such topics of articles.
2. The abstract should be rewritten: it should not contain abbreviations and should be readable and understandable without reading the paper.
3. Keywords: 13DC – what is this? In my opinion, this abbreviation should be replaced with a whole phrase, since 13DC is not a standard abbreviation.
4. «The best known reaction to obtain the pyrazole and oxadiazole derivatives is by 1,3-dipolar cycloaddition reactions» there are lots of ways to prepare these heterocycles, and in my opinion, 1,3-DC is not the best one. This sentence should be revised and explained.
5. The introduction is redundant and should either be redone or more clearly structured and illustrated.
6. 2.3 should be named Chemistry or Experimental chemistry or Synthesis of …
7. Scheme 2: why stereobonds are used? The compounds are not chiral.
8. «N.M.R.» should be «NMR»
9. «I.R spectra» should be «IR spectra»
10. «electronic impact (E.I.)» should be «electronic impact (EI)»
11. Compound (D1Cl) was reported earlier (Hamdi, N.; Khemiss, A. Journal de la Societe Algerienne de Chimie (2002), 12(1), 45-52), and thus, a corresponding reference is required along with a comparison of earlier reported m.p. and the m.p. from the present paper.
12. 2.3.1-2.3.9 – all compounds are named incorrectly, MeO is para-, not ortho-.
13. In experimental section origin of all used reagents and solvents should be mentioned.
14. 2.3.1 should contain a reference for the synthesis of DAP. I.e. DAP was prepared by the method reported [00]. A similar description should be provided for chalcone derivative used.
15. For compounds 2.3.1-2.3.9 should be given a general synthetic procedure.
16. For compounds 2.3.1-2.3.9 signals of aromatic C in 13C NMR should be listed.
17. For each compound an elemental analysis or HRMS should be provided.
18. The manuscript must be provided with a supporting information, containing NMR plots for each new compound.
Author Response
Dear Editor of Molecules,
We are pleased to resubmit for publication the revised version Manuscript ID Molecules-2205941 entitled “DFT study of regio- and stereoselective [3+2] cycloaddition between diazopropane and substituted chalcone derivatives and molecular docking of novel pyrazole derivatives as anti-Alzheimer’s agents”. We appreciate the constructive insightful comments. We carefully examined your comments and made several amendments to the manuscript, which we hope meet with your approval. We replied your questions or comments in details in the following texts.
Reviewer 1
Comment 1: The topic should be changed: «:» should be replaced with «and». Or any other ideas. But «:» is not applicable for such topics of articles.
Reply: Thank you for your remark. We have modified the topic and taken into account your suggestion.
Comment 2: The abstract should be rewritten: it should not contain abbreviations and should be readable and understandable without reading the paper.
Reply: Thank you for your comment. We have rewritten the abstract in order to be readable and understandable without reading the paper.
Comment 3: Keywords: 13DC – what is this? In my opinion, this abbreviation should be replaced with a whole phrase, since 13DC is not a standard abbreviation.
Reply: Thank you for your comment. The 32CA keyword is commonly used for this type of 3+2 cycloaddition reaction.
Comment 4: «The best known reaction to obtain the pyrazole and oxadiazole derivatives is by 1,3-dipolar cycloaddition reactions» there are lots of ways to prepare these heterocycles, and in my opinion, 1,3-DC is not the best one. This sentence should be revised and explained.
Reply: Thank you for your remark. This sentence was modified and we have cited the usefulness of the [3+2] cycloaddition reaction in order to synthesize the pyrazole and oxadiazole derivatives.
Comment 5: The introduction is redundant and should either be redone or more clearly structured and illustrated.
Reply: Thank you for your suggestion. The introduction was revised and we have rewritten many parts in order to more clarify the topic and the objectives of the paper.
Comment 6: 2.3 should be named Chemistry or Experimental chemistry or Synthesis of …
Reply: Thank you for your comment. The 2.3 section caption was modified as Synthesis of pyrazole derivatives (D1-D3)
Comment 7: Scheme 2: why stereobonds are used? The compounds are not chiral.
Reply: Thank you for your remark. We have rectified this mistake in scheme 2.
Comment 8: «N.M.R.» should be «NMR»
Reply: Thank you for your remark. We have rectified this mistake.
Comment 9: «I.R spectra» should be «IR spectra»
Reply: Thank you for your remark. We have rectified this mistake.
Comment 10: «electronic impact (E.I.)» should be «electronic impact (EI)»
Reply: Thank you for your remark. We have rectified this mistake.
Comment 11: Compound (D1Cl) was reported earlier (Hamdi, N.; Khemiss, A. Journal de la Societe Algerienne de Chimie (2002), 12(1), 45-52), and thus, a corresponding reference is required along with a comparison of earlier reported m.p. and the m.p. from the present paper.
Reply: Thank you for your remark. We compared our melting point values in the case D1Cl with those reported by N. Hamdi et al., 2002) and close values were obtained.
Comment 12: 2.3.1-2.3.9 – all compounds are named incorrectly, MeO is para-, not ortho-.
Reply: Thank you for your remark. We have corrected the nomenclature of all compounds.
Comment 13: In experimental section origin of all used reagents and solvents should be mentioned.
Reply: Thank you for your question. The origin of all used reagents and solvents is Sigma Aldrich and this is indicated in experimental section.
Comment 14: 2.3.1 should contain a reference for the synthesis of DAP. I.e. DAP was prepared by the method reported [00]. A similar description should be provided for chalcone derivative used.
Reply: Thank you for your suggestion. For the synthesis of diazopropane, the work of D. E aplequist and H. Babad (J. Org. Chem. 1962, 27, 288–290) was cited. We have also included the work of Bolte et al., who reported the synthesis of methoxychalcone including crystallographic and experimental details.
Comment 15: For compounds 2.3.1-2.3.9 should be given a general synthetic procedure.
Reply: Thank you for your suggestion. The synthetic procedure of different compound was detailed in Sections 2.3.1, 2.3.5 and 2.3.9.
Comment 16: For compounds 2.3.1-2.3.9 signals of aromatic C in 13C NMR should be listed.
Reply: Thank you for your remark. We have indicated signals of aromatic 13C NMR in the section 2.3: Synthesis of pyrazole derivatives (D1-D3).
Comment 17: For each compound an elemental analysis or HRMS should be provided.
Reply: Thank you for your suggestion. We preceded MS spectra of the synthesized product and we regret to perform these analyses because our institution doesn’t possess the corresponding materials.
Comment 18: The manuscript must be provided with a supporting information, containing NMR plots for each new compound.
Reply: Thank you for your suggestion. NMR 1D, 2D and MS were included in the supporting information.

Reviewer 2 Report
(1) According to the actual state of the knowledge [Eur. J. Org. Chem. 267–282 (2019)], not all compounds considered earlier as 1,3-dipoles exhibit dipolar nature. So, some works should be replaced as follow:
- "1,3-dipole" - "three atom 4pi-component"
- "dipolarophile" - "2pi-component"
- "1,3-dipolar cycloaddition" - "[3+2] cycloaddition"
- "13DC" - "32CA"
etc. etc. etc. ...
(2) Due to issues mentioned above, the data presented on the scheme 1 are completly outdated. This scheme should be removed.
(3) At this moment several mechanisms should be considered regarding to 32CA reactions:
- Non-polar mechanisms (synchronical mechanism or stepwise, biradical mechanism [for example: Helv.Chim.Acta., v98, p453 (2015)])
- Polar mechanisms (one step-two stage mechanism [for example: Journal of Cleaner Production , 292, 126079 (2021)],
stepwise zwitterionic mechanism [for example: Computational and Theoretical Chemistry, 1125, 77-85 (2018); RSC Advances, 5, 101045-101048 (2015); Tetrahedron Letters, 56, 532 (2015) ]).
Additionaly, zwitterionic or biradical adducts with "extended conformation" may be exist in reaction environment independently of 3+2 cycloadducts [Monatsh.Chem. v146, p591 (2015)]
These issues should be mentioned.
(4) Application of local Fukui indices as well as Softness parameters are completly outdated. For this purpose, local Parr reactivity descriptors should be used [RSC Advances 3, 1486–1494 (2013)].
(5) Clear procedure for cycloadditions with the participation of diazopropane should be added.
(6) According to the acual state of knowledge, the FMO interactions analysis is completly failed, for the prediction of the reactivity of 32CAs components [Molecules v21, 1319 (2016). For this purpose, the analysis of global aln local electrophilicity/electrophillicities are dediceted [Eur. J. Org. Chem. 1107 (2018)]. So, paragraph 3.2 and 3.3 should be removed.
(7) Figure 6: basis set should be specified.
(8) IRC plots should be transferred ot the Supplementary Material
(9) Reference 47 is not correct source for "l" indice.
Author Response
Dear Editor of Molecules,
We are pleased to resubmit for publication the revised version Manuscript ID Molecules-2205941 entitled “DFT study of regio- and stereoselective [3+2] cycloaddition between diazopropane and substituted chalcone derivatives and molecular docking of novel pyrazole derivatives as anti-Alzheimer’s agents”. We appreciate the constructive insightful comments. We carefully examined your comments and made several amendments to the manuscript, which we hope meet with your approval. We replied your questions or comments in details in the following texts.
Reviewer 2
Comment 1: According to the actual state of the knowledge [Eur. J. Org. Chem. 267–282 (2019)], not all compounds considered earlier as 1,3-dipoles exhibit dipolar nature. So, some works should be replaced as follow:
- "1,3-dipole" - "three atom 4pi-component"
- "dipolarophile" - "2pi-component"
- "1,3-dipolar cycloaddition" - "[3+2] cycloaddition"
- "13DC" - "32CA" etc. etc. etc. ...
Reply: Thank you for your suggestion. We have rectified these terms and abbreviations as recommended.
Comment 2: Due to issues mentioned above, the data presented on the scheme 1 are completly outdated. This scheme should be removed.
Reply: Thank you for your remark. We have removed scheme 1 as recommended.
Comment 3: At this moment several mechanisms should be considered regarding to 32CA reactions:
- Non-polar mechanisms (synchronical mechanism or stepwise, biradical mechanism [for example: Helv.Chim.Acta., v98, p453 (2015)])
- Polar mechanisms (one step-two stage mechanism [for example: Journal of Cleaner Production, 292, 126079 (2021)],stepwise zwitterionic mechanism [for example: Computational and Theoretical Chemistry, 1125, 77-85 (2018); RSC Advances, 5, 101045-101048 (2015); Tetrahedron Letters, 56, 532 (2015) ]). Additionaly, zwitterionic or biradical adducts with "extended conformation" may be exist in reaction environment independently of 3+2 cycloadducts [Monatsh. Chem. v146, p591 (2015)]
These issues should be mentioned.
Reply: Thank you for suggestion. We have taken into account your suggestion and these data are included in the introduction section.
Comment 4: Application of local Fukui indices as well as Softness parameters are completly outdated. For this purpose, local Parr reactivity descriptors should be used [RSC Advances 3, 1486–1494 (2013)].
Reply: Thank you for your remark. We have taken into account the suggestion of the reviewers and we have computed the local Parr reactivity descriptors and results are added in the text (Table 1).
Comment 5: Clear procedure for cycloadditions with the participation of diazopropane should be added.
Reply: Thank you for your suggestion. A clear mechanism of the [3+2] cycloaddition of diploarophile (chalcone derivative) and dipole (diazopropane) is illustrated in Figure 1.
Comment 6: According to the actual state of knowledge, the FMO interactions analysis is completly failed, for the prediction of the reactivity of 32CAs components [Molecules v21, 1319 (2016). For this purpose, the analysis of global aln local electrophilicity/electrophillicities are dediceted [Eur. J. Org. Chem. 1107 (2018)]. So, paragraph 3.2 and 3.3 should be removed.
Reply: Thank you for your comment. We agree with the reviewers. Paragraph 3.2 was removed and section 3.3 was modified by analyzing Parr local reactivity descriptors and molecular orbitals coefficients in order to investigate the local reactivity of the studied reaction.
Comment 7: Figure 6: basis set should be specified.
Reply: Thank you for your comment. The basis was indicated in the figure caption.
Comment 8: IRC plots should be transferred ot the Supplementary Material.
Reply: Thank you for your suggestion. The IRC plots were transferred on the supporting informations.
Comment 9: Reference 47 is not correct source for "l" indice.
Reply: Thank you for your comment. This reference was removed.
PS: Any changes in the main document or in the supporting informations are highlighted in yellow.
Again, we appreciate all your insightful and constructive comments. We have asked for all the questions and remarks. We have corrected all the grammar mistakes. We really hope that these modifications can meet with your approval. Thank you very much.
Yours Sincerely,
Pr. Naceur HAMDI

Round 2
Reviewer 1 Report
The authors have made all needed changes; the manuscript is ready for acceptance.
Reviewer 2 Report
Authors considered all remarks and improved the manuscript accordingly. So, i recommend this paper for the further evaluation in Molecules journal.